# Illness anxiety disorder and somatic symptom disorder: Similarities and differences in health-anxious individuals

Katarina Kikas[1,2], Emily Upton[1,2], Brittany Corkish[1], Aliza Werner-Seidler[1,2], Jill M. Newby[1,2]*

1 Black Dog Institute, University of New South Wales, Sydney, New South Wales, Australia, 2 School of Psychology, University of New South Wales, Sydney, New South Wales, Australia

* j.newby@unsw.edu.au

## Abstract

Little is known about the features and characteristics of illness anxiety disorder (IAD) and somatic symptom disorder (SSD), and whether the minimal somatic symptom requirement in IAD is valid and useful. This study compares IAD and SSD, and IAD (involving no or mild somatic symptoms) to a modified IAD diagnosis (modified to include moderate-to-severe somatic symptoms) in health anxious individuals. We recruited health anxious individuals drawn from the community and assessed IAD and SSD. We compared the disorders on demographic and clinical characteristics, health care utilization, and assessed the prevalence of IAD subtypes. The validity of the IAD somatic symptom criterion was examined by comparing current IAD with modified IAD. Among 118 participants, 28 met criteria for IAD, 47 for SSD, and 38 for modified IAD. Most with IAD fluctuated between seeking and avoiding medical care (71.4%), while 25% were care-seeking and 3.6% were care-avoidant. Demographic information, illness course, mental health comorbidities, and symptom severity did not differ significantly between IAD and SSD. However, individuals with SSD reported significantly more somatic symptoms, chronic health conditions, reduced quality of life, and more health care visits compared to individuals with IAD. Similar findings were observed between IAD-current and IAD-modified. The minimal differences between IAD and SSD, and between IAD-current and IAD-modified call into question the utility of distinguishing health anxiety presentations based on somatic symptom presence and severity. Further research should compare these disorders on illness course and treatment outcomes.

## Introduction

Health anxiety in its severe form is a persistent condition marked by excessive fear of developing or having a serious illness or disease. In the DSM-5, it is the main

**Data availability statement:** Data cannot be shared publicly because our data contains highly sensitive mental health information. Data will be made available upon request only and subject to approval from our ethics committee (HREC number HC220649, contact via humanethics@unsw.edu.au). Data will be made available for researchers who meet criteria for access to confidential data.

**Funding:** This research was funded by the Australian National Health and Medical Research Council Investigator Grant scheme awarded to J. Newby (#2008839). The funders had no role in the study design, data collection and analysis, decision to publish, or preparation of the manuscript.

**Competing interests:** Jill Newby receives grant funding from the NHMRC, Wellcome Trust, MRFF, NIHR, and KONESKI. Aliza Werner-Seidler receives grant funding from NHMRC and MRFF. This does not alter our adherence to PLOS ONE policies on sharing data and materials.

characteristic of a diagnosed mental health disorder, known as illness anxiety disorder [1, 2]. Severe health anxiety has a debilitating impact at both the individual and societal level. Individuals who experience severe health anxiety report lowered quality of life, interference with everyday tasks (i.e., household duties, self-care, and mobility), and strained personal relationships and social roles [3, 4]. Moreover, at a societal level, one study [5] demonstrated that those with severe health anxiety reported increased absenteeism from work due to sick leave and in another study [2], health anxious individuals were found to utilize health care services (i.e., primary care, medication, psychiatric visits, general hospital care) at double the rate of non-health-anxious individuals with medical conditions.

The DSM-5 introduced two new diagnostic entities that can be used to categorize individuals with severe health anxiety: illness anxiety disorder (IAD) and somatic symptom disorder (SSD) [1]. For these individuals, both disorders involve concerns around the development and/or presence of disease. However, the key distinguishing factor between the disorders in clinical-level health anxiety is the presence and severity of somatic symptoms. Individuals with IAD experience persistent and excessive worry about having or developing a serious illness (i.e., cancer, heart disease) lasting at least six months, coupled with excessive health-related behaviors (i.e., repeated health-related googling, reassurance-seeking) or avoidance (i.e., avoiding medical settings), in the absence of somatic symptoms, or only mild ones if present. In contrast, individuals with SSD experience one or more severe and persistent somatic symptoms that significantly disrupt their daily life, along with high levels of anxiety around their health and physical symptoms. The current study focuses on individuals with elevated health anxiety, including people with IAD and the subpopulation of people with SSD who experience severe and debilitating health anxiety. We acknowledge that it is possible to meet criteria for SSD without experiencing health anxiety, such as those individuals that are debilitated by their persistent physical symptoms (i.e., chronic pain, chronic fatigue) [1].

Given that these diagnostic entities were introduced in the most recent DSM-5 as a new way to diagnose health anxiety, research is needed to evaluate the validity and clinical utility of distinguishing them as separate disorders. To date, three studies have examined the diagnostic validity and clinical utility of IAD and SSD in health-anxious individuals [6–8]. Bailer et al. [7] were the first to examine the clinical utility of distinguishing IAD and SSD on somatic symptoms using a 'probable' IAD and SSD diagnosis, which was operationalized from self-report questionnaires and clinical interviews. Specifically, in a sample of treatment-seeking health anxious individuals, the authors examined whether the disorders differed on various qualitative characteristics, including cognitive (i.e., rumination about physical symptoms or illnesses), affective (i.e., health anxiety), and behavioral features (i.e., doctor shopping). The findings revealed more similarities than differences between IAD and SSD. With respect to the observed differences, the authors found that health-anxious individuals with SSD exhibited more severe symptoms and higher healthcare utilization rates than health-anxious individuals with IAD. Specifically, individuals with SSD were more functionality impaired, experienced a higher number of somatic symptoms,

had greater comorbidity with other anxiety disorders (i.e., panic disorder and generalized anxiety disorder), and attended health services more frequently than health anxious individuals with IAD. These findings led the authors to conclude that SSD is a more severe disorder than IAD in individuals with severe health anxiety and that the distinctions between the disorders were a matter of severity rather than of any qualitative differences. However, a limitation of this study was that IAD and SSD were operationalized using a post-hoc definition of the disorders, based on information gathered from self-report questionnaires and clinical interviews (for DSM-IV diagnoses) rather than being based on DSM-5 diagnostic criteria.

To address this gap, Newby et al. [6] replicated and extended Bailer et al. [7] study by examining the validity and clinical usefulness of DSM-5 IAD and SSD diagnoses in a cohort of individuals with health anxiety who signed up to participate in a treatment-trial. Consistent with Bailer et al. [7], they found more similarities than differences between IAD and SSD. The observed differences between IAD and SSD were related to symptom severity and mental health comorbidities, with the SSD group reporting greater somatic symptom severity, depression severity, functional impairment and higher health care utilization compared to the IAD group. These findings led the authors to conclude that differences between the disorders relate to severity rather than qualitative differences.

More recently, a Swedish team aimed to replicate these results while accounting for trait health anxiety in a treatment-trial sample [6]. After controlling for trait health anxiety, they found no phenomenological differences in demographic and clinical characteristics related to health anxiety (e.g., age, gender, education, employment, anxiety sensitivity, intolerance to uncertainty, avoidance and safety behaviors), clinical course (e.g., age of onset) and visits to general practitioners or medical specialists. However, the authors found that SSD was associated with a greater number of psychologist visits, higher levels of disability, and more pain and tinnitus than IAD. These findings suggest that although minor differences between IAD and SSD remain after accounting for trait health anxiety, the overall distinction between the two diagnoses appear to have limited clinical value.

Taken together, the findings from these three studies suggest that there is little distinction between IAD and SSD in individuals with health anxiety in terms of cognitive, behavioral, and affective features. The similarities between these two disorders suggests limited clinical utility of splitting health anxiety into two separate disorders. However, as these studies were conducted with treatment-seeking samples, further research is needed comparing IAD and SSD in non-treatment-seeking samples, such as individuals drawn from the community.

In the DSM-5, Criterion B for IAD specifies that health anxious individuals must have either no somatic symptoms or only mild ones, setting them apart from individuals who meet criteria for SSD who typically experience more pronounced physical symptoms. However, some researchers have questioned whether this strict IAD criteria is necessary and clinically useful [9], as clinical observations suggest that that some individuals with IAD present with moderate to severe somatic symptoms [9]. To date, no empirical studies have directly tested the validity of IAD Criterion B. Research is needed to compare individuals who meet current IAD criteria (i.e., no or minimal somatic symptoms) to those who meet all IAD criteria but report moderate-to-severe somatic symptoms. Comparing these groups is crucial for evaluating the validity of criterion B in the diagnosis of IAD. It will help to determine whether this criterion accurately captures how the disorder presents in real-world clinical settings or needs to be revised. The primary objective of this study was to replicate and extend previous research [6–8] in a sample of individuals drawn from the community who self-identified as experiencing anxiety about their health. We aimed to examine: (1) the validity and clinical utility of having two separate disorders for health anxiety based on somatic symptom presence and severity, and (2) the validity of IAD Criterion B. Specifically, the first aim was to investigate the features and characteristics of IAD and SSD. To achieve this, we recruited individuals with self-reported health anxiety, conducted diagnostic assessments for IAD and SSD and compared those who met diagnostic criteria for IAD and SSD on various demographic characteristics (e.g., age, gender, employment, education), clinical (e.g., symptom severity, functional impairment, and mental and physical comorbidity) and behavioral characteristics (e.g., healthcare utilization). The second aim of this study was to test the validity of IAD Criterion B, by examining whether health-anxious individuals who met current IAD criteria (i.e., no or mild somatic symptoms) differed from those who met

all DSM-5 IAD criteria but reported experiencing moderate-to-severe somatic symptoms. We use the term 'modified IAD diagnosis' throughout this paper to describe the latter group.

In line with previous findings [6–8], we hypothesized that differences between IAD and SSD in individuals with health anxiety would be related to severity, with SSD individuals scoring higher on measures of generalized anxiety, health anxiety, depression, somatic symptom severity, functional impairment, and have more mental health comorbidities and greater health care utilization. Similarly, we hypothesized that individuals with a modified IAD diagnosis (moderate-to-severe somatic symptoms) would experience greater severity on clinical characteristics (i.e., symptom severity, functional impairment) and higher rates of healthcare utilization compared to those with the current IAD diagnosis (minimal or no somatic symptoms). We did not expect differences among the groups on demographic characteristics, features of illness anxiety, or health anxiety onset and course.

## Method

### Participants and procedure

One hundred and eighteen participants were recruited between November 2022 and 2023 through targeted online media advertisements on Instagram and Facebook. These advertisements were aimed at individuals who worried about their health or identified as having health anxiety, using messages such as "Are you suffering from health anxiety?" and "We are seeking research participants who experience persistent and frequent worries about their health…". Interested individuals were directed to the Black Dog Institute website, which outlined the eligibility criteria and provided a link to a Qualtrics (2022) questionnaire. The questionnaire assessed eligibility, and eligible participants received an electronic participant information statement and consent form. Eligible participants were aged over 18, Australian residents, fluent in English, and self-identified as experiencing persistent and frequent concerns and worries about their health. The study received approval from the University of New South Wales Human Research Ethics Committee (HC220649), and all participants provided electronic informed consent to participate.

After providing consent, participants completed a 45-minute online questionnaire and were invited to participate in an online interview via Zoom Videoconferencing (2016), ranging between 15–60 minutes in duration. The purpose of the interview was to assess diagnostic status for IAD, SSD and comorbid disorders including generalized anxiety disorder, obsessive and compulsive disorder, panic disorder, agoraphobia, and major depressive disorder. Participants were reimbursed at a rate of $35 per hour through an electronic gift-card upon the completion of both the survey and interview.

### Measures

**Self-report measures.** Participants completed a battery of questionnaires, including demographic questions, symptom severity measures, quality of life, and health service use in the past three months (detailed below). The demographic questionnaire included age, gender, ethnicity, residential suburb, place of birth, primary language at home, employment and relationship status, and level of education. Additionally, participants provided information about their past and current general mental health treatment (i.e., that was not specific to health anxiety) and current chronic health conditions.

The Illness Anxiety Questionnaire [6] measured health anxiety onset, course, and characteristics, such as care-seeking, care-avoidant, and fluctuating subtypes. Symptom severity measures included the Short Health Anxiety Inventory [SHAI; 10, $\alpha = .88$] for assessing health anxiety severity, the Patient Health Questionnaire [PHQ-9; 11, $\alpha = .88$] for measuring depression severity, the Patient Health Questionnaire Somatic Symptom Severity Scale [PHQ-15; 12, $\alpha = .71$] for somatic symptom severity, and the Generalized Anxiety Disorder Assessment scale [GAD-7; 13, $\alpha = .9$] for anxiety.

The ten-item Recovering Quality of Life scale [ReQol-10; 14, $\alpha = 0.85$] assessed the quality of life for people experiencing mental health difficulties over the last week, and the Short-Form Health Survey [SF-12; 15, $\alpha = 0.85$] assessed the impact of health on everyday life. Participants answered a range of questions regarding their use of health care services

(i.e., GP, psychologist, psychiatrist, medical specialist visits) over the past three months, assessed via questions adapted from the service use module from the CIDI 2.0 [16].

**Diagnostic interview.** An abbreviated version of the Anxiety and Related Disorders Interview Schedule for DSM-5 -Adult Version [ADIS-V; 17] was used to assess current DSM-5 IAD or modified IAD diagnosis, SSD, major depressive disorder, panic disorder, generalized anxiety disorder, obsessive compulsive disorder, and agoraphobia. To receive an IAD modified diagnosis, a question was included in the IAD section that asked participants to rate the severity of their current physical symptoms (i.e., "How severe are these physical symptoms? Not at all, mild, moderate, moderately severe, severe"). Participants that rated their physical symptoms as 'not at all severe' or 'mild' received the current IAD diagnosis. If symptoms were rated above the moderate threshold, participants received the IAD modified diagnosis if they met all remaining IAD criteria. The ADIS-V was administered by researchers KK and clinical psychologists EU and BC. EU and BC, already trained in administering the ADIS-V, provided training to KK. Consistency was maintained by adhering strictly to the structured interview format, using the exact wording of the script without deviation. We also assessed interrater reliability via blind ratings of audio recordings (see section below). Regular supervision with clinical psychologists and researchers AWS and JN was conducted to discuss any cases that required clarification, discussion and/or an additional rating.

## Statistical analyses

**Validity of DSM-5 IAD, SSD and modified IAD.** All analyses were performed with SPSS (Statistical Packages for Social Sciences) Version 27. Means were computed for continuous variables, and frequencies were calculated for the categorical variables. Independent *t*-tests were conducted to compare means between IAD and SSD, as well as between IAD current and IAD modified to assess differences in symptom severity, quality of life, and service utilization. Participants meeting comorbid IAD and SSD criteria were excluded from analyses comparing IAD and SSD to address the research question of whether differences exist between the two disorders. Chi square analyses were used to compare groups on categorical variables, such as demographic characteristics, onset, course, and mental health comorbidities. Cohen's *d* was calculated to examine effect sizes, and binary logistic regression was employed to calculate odds ratios. A $p < 0.05$ was considered as statistically significant.

**Inter-rater reliability.** To assess inter-rater reliability, all interviews were recorded, and participants meeting IAD or SSD criteria were assessed by an independent blinded rater. A subset of interviews were coded for the remaining diagnoses (e.g., MDD, 10%; n = 18). Two clinical psychologists (BC, MH) listened to the recordings and made an independent diagnosis; these raters were blinded to the diagnosis recorded by the initial interviewer. Any discrepancies between the raters were resolved by a third rater (EU/JN).

## Results

A total of 386 individuals consented to participate in the study via the online survey. Among them, 124 did not complete the survey, resulting in 262 participants who fully completed the survey. Of the 262, 118 completed the diagnostic interviews and were included in the final sample (see Table 1 for Participant Characteristics).

### Sample demographic characteristics and associations between DSM-5 IAD and SSD

Demographic results for the entire sample can be found in S1 Table, whereas demographic information for IAD and SSD are presented in Table 1. The majority of the sample identified as female (n = 99, 83.9%), had a mean age of 41 (range = 18–72 years, SD = 12.4), and were of Australian ethnicity (n = 74, 69.2%) and born in Australia (n = 86, 72.9%). The sample primarily spoke English at home (n = 105, 89%) and resided in major cities in Australia (n = 89, 75.4%). Approximately half of the sample were in a de facto or married relationship (n = 58, 49.2%), and more than a half were employed part-time or full-time (n = 76, 64.4%) and had completed an undergraduate degree (n = 47, 39.8%) or postgraduate degree (n = 32, 27.1%). Most of the sample (n = 105,

**Table 1. Demographic characteristics of participants with IAD and SSD.**

| | IAD (n = 28) | SSD (n = 47) | IAD vs SSD | |
|---|---|---|---|---|
| | n (%) | n (%) | Statistic | OR (95%CI) |
| Gender | | | χ² (2) = 2.82, p = 0.24 | |
| Man or male | 4 (14.3) | 2 (4.3) | | 0.50 (0.04-6.68) |
| Woman or female | 22 (78.6) | 43 (91.5) | | 1.96 (0.26-14.86) |
| Non-binary or different term | 2 (7.2) | 2 (4.3) | | 1.00 |
| Ethnicity | | | χ² (2) = 0.80, p = 0.37 | |
| Australian | 19 (67.9) | 27 (57.4) | | 0.64 (0.24-1.71) |
| Other | 9 (32.1) | 20 (42.6) | | 1.0 |
| Birthplace Australia | 22 (78.6) | 35 (74.5) | χ² (1) = 0.0, p = 0.96 | 0.80 (0.26-2.43) |
| English primary language at home | 25 (89.3) | 44 (93.6) | χ² (1) = 1.0, p = 0.75 | 1.76 (0.33-9.39) |
| Residence in Australia | | | χ² (1) = 2.28, p = 0.13 | |
| Major cities/urban | 23 (82.1) | 31 (66.0) | | 0.42 (0.14-1.32) |
| Regional or remote | 5 (17.9) | 16 (34.0) | | |
| Relationship status | | | χ² (3) = 7.32, p = 0.06 | |
| Single | 7 (25) | 14 (29.8) | | 2.0 (0.66-6.01) |
| De facto/Married | 19 (67.9) | 19 (40.4) | | 1.00 |
| Divorced/Separated/Widowed | 2 (7.2) | 10 (21.3) | | 5.0 (0.96-25.93) |
| Other (i.e., partnered living apart and solo-polyamorous) | 0 (0) | 4 (8.5) | | – |
| Employment status | | | | |
| Unemployed | 5 (17.9) | 6 (12.8) | χ² (1) = 0.36, p = 0.55 | 0.67 (0.19-2.45) |
| Employed full-time | 13 (46.4) | 6 (12.8) | χ² (1) = 10.51, p < 0.05 | 0.17 (0.05-0.53) |
| Employed part-time | 8 (28.6) | 21 (44.7) | χ² (1) = 1.92, p = 0.17 | 2.02 (0.74-5.50) |
| Stay-at-home parent | 2 (7.1) | 2 (4.3) | χ² (1) = 0.29, p = 0.59 | 0.58 (0.78-4.35) |
| Carer for family member (not children) | 1 (3.6) | 1 (2.1) | χ² (1) = 0.14, p = 0.71 | 0.59 (0.04-9.77) |
| Other (i.e., student, casual work, disability pension, retirement) | 1 (3.6) | 14 (29.8) | χ² (1) = 7.54, p < 0.05 | 11.46 (1.42-92.75) |
| Level of education | | | χ² (3) = 5.87, p = 0.12 | |
| High school level | 2 (7.1) | 6 (12.8) | | 1.0 |
| Certificate/diploma | 9 (32.1) | 10 (21.3) | | 0.37 (0.06-2.32) |
| University undergraduate degree | 8 (28.6) | 24 (51.1) | | 1.0 (0.17-5.99) |
| University postgraduate degree | 9 (32.1) | 7 (14.9) | | 0.30 (0.04-1.70) |
| Past general mental health treatment | 25 (89.3) | 42 (89.4) | χ² (1) = 0.0, p = 0.99 | 1.01 (0.22-4.58) |
| Past treatment type | | | | |
| Medication | 16 (59.3) | 35 (83.3) | χ² (1) = 5.94, p < 0.05 | 3.44 (1.13-10.51) |
| Therapy with psychologist | 23 (85.2) | 40 (95.2) | χ² (1) = 2.10, p = 0.15 | 3.48 (0.59-20.49) |
| Therapy with psychiatrist | 7 (25.9) | 16 (38.1) | χ² (1) = 1.10, p = 0.30 | 1.76 (0.61-5.09) |
| Support from GP | 16 (59.3) | 27 (64.3) | χ² (1) = 0.18, p = 0.67 | 1.24 (0.46-3.34) |
| Counselling from other mental health professional (i.e., nurse, social worker) | 10 (37) | 14 (33.3) | χ² (1) = 0.10, p = 0.75 | 0.85 (0.31-2.34) |
| Online mental health program | 5 (18.5) | 13 (31.0) | χ² (1) = 1.32, p = 0.25 | 1.97 (0.61-6.36) |
| Over-the-counter medication (e.g., vitamins) | 3 (11.1) | 12 (28.6) | χ² (1) = 2.95, p = 0.09 | 3.20 (0.81-12.65) |
| Other (i.e., exercise, diet changes, ECT) | 2 (7.4) | 2 (4.8) | χ² (1) = 0.21, p = 0.646 | 0.63 (0.08-4.72) |
| Current general mental health treatment | 15 (53.6) | 30 (63.8) | χ² (1) = 0.77, p = 0.38 | 1.53 (0.59-3.96) |
| Current treatment type | | | | |
| Medication | 11 (50) | 21 (70.0) | χ² (1) = 2.15, p = 0.14 | 2.33 (0.74-7.32) |
| Therapy with psychologist | 9 (40.9) | 22 (73.3) | χ² (1) = 5.54, p < 0.05 | 3.97 (1.23-12.84) |
| Therapy with psychiatrist | 1 (4.5) | 4 (13.3) | χ² (1) = 1.13, p = 0.29 | 3.23 (0.34-31.13) |

*(Continued)*

**Table 1.** (Continued)

| | IAD (n=28) | SSD (n=47) | IAD vs SSD | |
|---|---|---|---|---|
| Support from GP | 7 (31.8) | 16 (53.3) | $\chi^2$ (1) = 2.38, p=0.12 | 2.45 (0.78-7.72) |
| Counselling from other mental health professional (i.e., nurse, social worker) | 3 (13.6) | 5 (16.7) | $\chi^2$ (1) = 0.09, p=0.77 | 1.27 (0.27-5.97) |
| Online mental health program | 3 (13.6) | 1 (3.3) | $\chi^2$ (1) = 1.90, p=0.17 | 0.22 (0.02-2.26) |
| Over-the-counter medication (e.g., vitamins) | 6 (27.3) | 7 (23.3) | $\chi^2$ (1) = 0.12, p=0.75 | 0.81 (0.23-2.87) |
| Other (i.e., exercise, diet changes, ECT) | 1 (4.5) | 3 (10.0) | $\chi^2$ (1) = 0.53, p=0.47 | 2.33 (0.23-24.01) |

IAD; Illness anxiety disorder without comorbidity of other listed disorders, SSD; Somatic Symptom Disorder without comorbidity of other listed disorders, OD; Odds ratio; 95%(CI); 95% Confidence Intervals, 1.0; reference category.

89%) had sought treatment for their mental health in the past, with just over a half (n=67, 56.8%) currently receiving mental health treatment. Of those currently receiving treatment, 60.8% (n=45) were undergoing therapy with a psychologist, 60.8% (n=45) were taking medication, and 45.9% (n=34) were receiving mental health support from their general practitioner.

## Prevalence of DSM-5 IAD and SSD

Inter-rater reliability was assessed for IAD, SSD, and comorbid disorders. Kappa estimates between the two raters were 0.96 for IAD, 0.88 for SSD. Discrepancies for IAD and SSD arose from disagreements regarding the severity of somatic symptoms. Inter-rater agreement for other comorbid diagnoses was 100% except for obsessive compulsive disorder (kappa=0.73).

Approximately one in four participants met criteria for IAD (n=28, 23.7%), while just under 40% met criteria for SSD (n=47, 39.8%). Comorbid IAD and SSD were present in 11 participants (9.3%), and 32 participants (27.1%) entered the study with self-identified health anxiety but did not meet criteria for either IAD (current or modified) or SSD, which was mostly due to participants not meeting the 6-month duration criteria for the disorders.

Among participants meeting IAD criteria, the majority self-identified as fluctuating between seeking and avoiding medical care (n=20, 71.4%), followed by 7 (25%) indicating they were care-seeking, and 1 (3.6%) indicating they were care-avoidant, as illustrated in Table 2. Please note that we do not report the care-seeking and care-avoidance results for SSD, as SSD in the DSM-5 does not contain these subtypes.

Participants meeting criteria for SSD differed from those with IAD in terms of significantly lower levels of fulltime employment, higher levels of casual work, more time off work, retirement and disability pension, greater past use of medication for mental health treatment, and recipients of greater levels of current psychological therapy sessions.

## Illness onset, nature and course of health anxiety

Illness nature and course for the overall sample are depicted in S2 Table and in Table 2 for IAD and SSD. Overall, the average age of health anxiety onset was 24 years (range 4–65; median 39). The majority of the sample (n=99, 83.9%) reported fears related to multiple illnesses, while 16.1% (n=19) reported fearing the same illness. The most frequently reported feared illnesses included cancer, multiple sclerosis, and heart disease. More than a half of the participants (n=65, 55.1%) reported experiencing greater than 7 distinct episodes of health anxiety lasting two or more weeks in their lifetime, and 56.8% (n=67) reported feeling anxious about their health for more than four years in their lifetime. Comparisons between participants meeting criteria for IAD and SSD showed no differences in terms of illness onset and course.

## Symptom severity, quality of life, and health care service utilization

As outlined in Table 3, the IAD and SSD groups did not show significant differences on health anxiety severity (d=0.01), depression severity (d=0.45), generalized anxiety (d=0.19), general quality of life (d=0.27) and quality

**Table 2. Nature and course of health anxiety of individuals with IAD and SSD.**

| | IAD (n = 28) | SSD (n = 47) | IAD vs SSD | |
| --- | --- | --- | --- | --- |
| | n (%) | n (%) | Statistic | OR (95%CI) |
| Illness fears | | | χ2 (1) = 3.69, p = 0.06 | |
| Feared same illness | 1 (3.6) | 9 (19.1) | | 1.0 |
| Feared multiple illnesses | 27 (96.4) | 38 (80.9) | | 0.16 (0.02-1.31) |
| Number of episodes | | | χ2 (1) = 0.02, p = 0.88 | |
| 1-7 episodes | 13 (46.5) | 21 (44.7) | | 0.93 (0.36-2.38) |
| Greater than 7 episodes | 15 (53.6) | 26 (55.3) | | 1.0 |
| Total lifetime duration (health anxiety) | | | χ2 (2) = 0.66, p = 0.72 | |
| < 2 years | 10 (35.7) | 13 (27.7) | | 1.0 |
| 2-4 years | 2 (7.1) | 5 (10.6) | | 1.92 (0.31-12.05) |
| Greater than 4 years | 16 (57.1) | 29 (61.7) | | 1.39 (0.50-3.89) |
| Illness anxiety subtype | | | χ2 (3) = 0.74, p = 0.86 | |
| Care-seeking subtype | 7 (25) | 13 (27.7) | | 0.93 (0.07-12.14) |
| Care-avoidant subtype | 1 (3.6) | 2 (4.3) | | 1.0 |
| Fluctuate between care- seeking and care-avoidance | 20 (71.4) | 31 (66.0) | | 0.78 (0.07-9.12) |
| None of the above | 0 (0.0) | 1 (2.1) | | – |

IAD; Illness anxiety disorder without comorbidity of other listed disorders, SSD; Somatic Symptom Disorder without comorbidity of other listed disorders, OD; Odds ratio; 95%(CI); 95% Confidence Intervals, 1.0; reference category.

**Table 3. Symptom severity, quality of life, and service utilization in individuals with IAD and SSD.**

| | IAD (n = 28) | SSD (n = 47) | IAD vs SSD | |
| --- | --- | --- | --- | --- |
| Symptom severity | M (SD) | M (SD) | Statistic | Effect size |
| Health anxiety (SHAI-18) | 30.3 (5.8) | 30.4 (7.4) | t(73) = 0.05, p = 0.96 | d = 0.01 |
| Somatic symptoms (PHQ-15) | 13.3 (4.5) | 15.9 (4.8) | t(73) = 2.36, p < 0.05 | d = 0.56 |
| Depression (PHQ-9) | 10.8 (6.3) | 13.7 (6.7) | t(73) = 1.89, p = 0.06 | d = 0.45 |
| Generalized anxiety (GAD-7) | 9.5 (5.03) | 10.5 (5.3) | t(73) = 0.81, p = 0.42 | d = 0.19 |
| Quality of life | | | | |
| REQoL-10 | 20.9 (6.8) | 19 (7.2) | t(73) = 1.14, p = 0.26 | d = 0.27 |
| SF-12 Mental | 35.3 (10.04) | 34.3 (8.7) | t(73) = 0.45, p = 0.66 | d = 0.12 |
| SF-12 Physical | 46.6 (9.2) | 34.2 (12.3) | t(73) = 4.6, p < 0.05 | d = 1.10 |
| | IAD (n = 28) | SSD (n = 45)* | IAD vs SSD | |
| Service Utilization | M (SD) | M (SD) | Statistic | Effect size |
| Total appointments | 5.6 (4.5) | 15.9 (17.0) | t(70) = 3.06, p < 0.05 | d = 0.75 |
| General practitioner | 2.2 (1.8) | 4.1 (4.1) | t(71) = 2.28, p < 0.05 | d = 0.55 |
| Psychologist | 1.1 (1.7) | 3.2 (4.4) | t(70) = 2.41, p < 0.05 | d = 0.59 |
| Psychiatrist | 0.2 (1.0) | 0.7 (2.4) | t(70) = 1.07, p = 0.29 | d = 0.26 |
| Medical specialists | 0.8 (1.6) | 1.8 (2.5) | t(71) = 1.92, p = 0.06 | d = 0.46 |
| Other health practitioners (i.e., physiotherapist, massage therapists, chiropractor) | 1.3 (2.4) | 6.1 (12.7) | t(71) = 1.97, p = 0.05 | d = 0.47 |

IAD; Illness anxiety disorder without comorbidity of other listed disorders, SSD; Somatic Symptom Disorder without comorbidity of other listed disorders, M; mean, SD; standard deviation, d; Cohen's d effect size.

*Note. Due to experimenter error, two participants in the SSD group did not complete the service utilization section of the survey (SSD; n = 45).

of life related to mental health ($d = 0.12$). Participants that met SSD criteria reported significantly higher levels of somatic symptoms ($d = 0.56$) and lower quality of life related to physical health compared to participants that met IAD criteria ($d = 1.1$). Participants with SSD reported significantly more health professional appointments in the past three months ($d = 0.75$), including more general practitioner appointments ($d = 0.55$) and psychologist appointments ($d = 0.59$) compared to participants meeting IAD criteria. However, there were no significant differences between groups in the number of psychiatrist appointments ($d = 0.26$), medical specialist appointments ($d = 0.46$), and other health practitioner appointments (i.e., physiotherapy, massage therapist, and chiropractors) ($d = 0.47$). Please note, due to an experimenter error three participants in the total sample did not complete the health service utilization section of the survey, with two being in the SSD group, bringing the total sample to 115 and SSD sample to 45 for this section. The symptom severity, quality of life, and health care service utilization results for the overall sample can be found in S3 Table.

## Mental and physical comorbidities

Mental and physical health comorbidities are presented in Table 4. Fifteen participants (53.6%) with IAD and 74 participants with SSD (62.7%) met criteria for other DSM-5 diagnoses that were assessed. There were no group differences

**Table 4. Health anxiety comorbidities with other mental health disorders and physical health conditions.**

| | Total sample (N = 118) | IAD (n = 28) | SSD (n = 47) | IAD vs SSD | |
|---|---|---|---|---|---|
| | n (%) | n (%) | n (%) | Statistic | OR (95%CI) |
| DSM-5 diagnoses | | | | | |
| Generalized anxiety disorder | 53 (44.9) | 10 (35.7) | 24 (51.1) | χ2 (1) = 1.67, p=0.20 | 1.88 (0.72-4.91) |
| Panic disorder | 15 (12.7) | 3 (10.7) | 8 (17.0) | χ2 (1) = 0.56, p=0.46 | 1.71 (0.41-7.06) |
| Agoraphobia | 23 (19.5) | 5 (17.9) | 11 (23.4) | χ2 (1) = 0.32, p=0.57 | 1.41 (0.43-4.57) |
| Obsessive compulsive disorder | 41 (34.7) | 6 (21.4) | 19 (40.4) | χ2 (1) = 2.85, p=0.09 | 2.49 (0.85-7.29) |
| Major depressive disorder | 34 (28.8) | 6 (21.4) | 19 (40.4) | χ2 (1) = 2.85, p=0.09 | 2.49 (0.85-7.29) |
| Somatic Symptom Disorder | 58 (49.2) | – | – | – | – |
| Illness anxiety disorder (current) | 39 (33.1) | – | – | – | – |
| Chronic illness | 56 (47.5) | 8 (28.6) | 33 (70.2) | χ2 (1) = 12.28, p<0.05 | 5.89 (2.10-16.52) |
| Current chronic illness | | | | | |
| Asthma | 15 (22.4) | 2 (10.5) | 9 (28.1) | – | – |
| Cancer | 3 (4.5) | 0 (0.0) | 2 (6.3) | – | – |
| Heart disease, stroke, or vascular disease | 5 (7.5) | 0 (0.0) | 3 (9.4) | – | – |
| Circulatory condition | 3 (4.5) | 0 (0.0) | 2 (6.3) | – | – |
| Muscular-skeletal disorders (Gout, rheumatism, osteoporosis or arthritis) | 14 (20.9) | 1 (5.3) | 9 (28.1) | – | – |
| Diabetes | 7 (10.4) | 2 (10.5) | 3 (9.4) | – | – |
| Back problems | 13 (19.4) | 1 (5.3) | 10 (31.3) | – | – |
| Chronic pain problems | 9 (13.4) | 1 (5.3) | 8 (25) | – | – |
| Autoimmune diseases | 10 (14.9) | 2 (10.5) | 6 (18.8) | – | – |
| Gynaecological disorders | 4 (6.0) | 0 (0.0) | 2 (6.3) | – | – |
| Other (i.e., eye conditions, connective tissue disorder, inflammatory disease) | 12 (17.6) | 0 (0.0) | 11 (33.3) | – | – |

IAD; Illness anxiety disorder without comorbidity of other listed disorders, SSD; Somatic Symptom Disorder without comorbidity of other listed disorders, OD; Odds ratio; 95%(CI); 95% Confidence Intervals, 1.0; reference category.

between IAD and SSD groups in terms of the proportion of those experiencing comorbid mental disorders. Participants meeting SSD criteria experienced muscular-skeletal disorders, back problems, and other conditions (i.e., bowel disease, eye conditions, and lung conditions) significantly more than participants meeting IAD criteria, but the groups did not differ on any of the other chronic health conditions that were assessed. Mental and physical comorbidities for the entire samples are reported in S4 Table.

## Discussion

The present study is the first to examine the features and characteristics of IAD and SSD in individuals drawn from the community who self-identified as experiencing health anxiety. We compared the two DSM-5 disorders on demographic and clinical characteristics, mental health comorbidities and health service use. Overall, we found limited differences between IAD and SSD in people with health anxiety, with differences primarily being related to severity. These results suggest that individuals with health anxiety who meet SSD criteria appear to be more severe in their presentation.

**Comparisons between IAD and IAD modified diagnostic criteria.** One third of the sample met criteria for a modified IAD diagnosis (i.e., IAD with moderate-to-severe somatic symptoms) (n = 38, 32.2%; see S5 Table). Participants with modified IAD reported greater amounts of casual work, were more likely to be in retirement and/or be on a disability pension and were more likely to be engaged in therapy with a psychiatrist compared to participants with IAD. Additionally, the modified IAD group experienced significantly higher levels of somatic symptoms ($d = 0.58$), poorer quality of life related to physical symptoms ($d = 0.70$), and more health professional appointments in the past three months ($d = 0.59$), including a higher frequency of general practitioner appointments ($d = 0.75$) compared to participants with current IAD. Furthermore, a majority of participants with modified IAD also met criteria for comorbid SSD and these individuals reported a greater number of conditions, such as POTS, bowel disease, eye conditions, and lung conditions, compared to participants meeting the current IAD criteria. There were no significant differences on other demographic characteristics, onset and course, symptom severity, quality of life, comorbidities, and health care utilization (See S6-S8 Table).

## Similarities and differences between IAD and SSD

Consistent with our previous research in a treatment-seeking sample who participated in a clinical trial for health anxiety treatment [6], we found no significant differences between IAD and SSD on most demographic characteristics and the nature and course of health anxiety, including number of health anxiety episodes, lifetime duration, subtypes and illness fears. However, we found that SSD was associated with a greater number of chronic health conditions compared to IAD. In contrast to our previous work, SSD was also associated with lower rates of full-time employment and greater medication use and therapy with psychologists than IAD. Given that individuals with SSD are debilitated by moderate-to-severe somatic symptoms, we speculate that the debilitating nature of these physical symptoms may prevent them from engaging in greater hours of work. However, it is unknown why SSD is associated with more medication use and therapy with psychologists. Although we did not find statistically significant differences between IAD and SSD on mental health comorbidities, numerically SSD had greater mental health comorbidities than IAD. Thus, it is possible that individuals with SSD need more psychological support and medication to manage these comorbidities.

In our sample, participants with health anxiety who met criteria for SSD had greater somatic symptom severity and poorer quality of life related to their physical health compared to participants with IAD, with medium to large effect sizes. These findings align with the broader literature on SSD indicating an association between SSD and reduced physical quality of life [18–20]. Past research in health-anxious samples has also found SSD to be significantly more comorbid with generalized anxiety disorder, major depressive disorder, panic disorder and agoraphobia, and associated with higher levels of health anxiety severity, depression severity, and functional impairment compared to IAD, with medium to large effect sizes [6, 7]. In contrast to this previous research, in the current sample of people from the community with health anxiety, we found non-significant small to medium differences between IAD and SSD on these variables. The reasons underlying these

discrepancies are unclear but may stem from sampling differences, as our study recruited health-anxious participants from the community, while past research recruited participants signing up to a treatment trial. It is possible that there may have been differences in severity or other factors which may have influenced the findings. Notably, however, we observed a numerical difference in depression severity, with greater depression severity in SSD that was approaching (but did not reach) statistical significance, indicating that our sample may have been underpowered to detect these differences between IAD and SSD.

Investigating healthcare utilization in individuals with health anxiety is important for developing targeted interventions that reduce unnecessary healthcare visits and promote symptom self-management when appropriate. Consistent with previous research [6], our results showed that participants who experienced health anxiety and met SSD criteria reported a greater total number of appointments with health professionals, as well as more visits to general practitioners and psychologists in the past three months compared to participants with IAD, with medium to large effect sizes. In addition, Newby et al. [6] found that SSD was associated with more frequent visits to psychiatrists and other health professionals, with a medium effect size. These findings may suggest that higher somatic symptom severity in SSD may prompt individuals to seek more medical care than those with no or milder somatic symptoms, such as in IAD. Although speculative, interventions aimed at reducing or managing somatic symptoms could help reduce unnecessary healthcare visits, thereby lessening the burden on both the individuals and the healthcare system.

### Illness anxiety nature, course, and subtypes

Very few studies have examined the nature of the newly identified IAD diagnosis in DSM-5 [21]. This study replicates our previous research in treatment-trial samples [6], and shows that health-anxious individuals drawn from the community experience IAD in a long and episodic course, fear multiple illnesses, and begin worrying about their health in early adulthood. Similarly, we found that the IAD care-avoidant subtype was rare, the care-seeking subtype was more common, and, interestingly, the majority of our sample fluctuated between seeking and avoiding medical care. These findings warrant further investigation into the features and characteristics of these subtypes, including why some individuals with IAD seek while others avoid medical care, as well as what prompts an individual to fluctuate between the two (i.e., a specific period in their lives, an unexpected somatic symptom, or their health anxiety severity). Such findings would call into question the necessity of categorizing individuals with IAD into two different subtypes, as they might be better suited as behavioral symptoms of IAD.

### Validity of somatic symptom severity criteria in IAD

There has been criticism regarding the clinical utility of the somatic symptom criterion (Criterion B) in DSM-5 IAD, particularly as clinical observations indicate that individuals with IAD can present with moderate-to-severe somatic symptoms [9]. This study is the first to test the validity of Criterion B by directly comparing individuals who meet current IAD criteria with those who meet modified IAD criteria (with moderate-to-severe somatic symptoms) across various demographic, clinical and behavioral characteristics. While some differences emerged (i.e., IAD modified group experienced greater casual employment history, past psychiatric care, poorer physical quality of life, more healthcare appointments than IAD current), the groups did not differ on many other demographic characteristics and clinical characteristics (i.e., illness anxiety course, symptom severity for generalized anxiety, health anxiety and depression, mental health comorbidities and chronic health conditions). This suggests that the presence or severity of somatic symptoms may be less important for clinicians when diagnosing an individual with IAD. More research is needed to examine the course and treatment response of IAD with or without somatic symptoms in various samples (i.e., treatment-seeking, primary and secondary care), as well as the role of differential DSM diagnostic criteria, such as Criterion F, in distinguishing IAD from related disorders including SSD.

### Limitations

The present study has several limitations that need to be addressed. First, although participants were drawn from the community rather than as a treatment-seeking sample, the majority (90%) had sought treatment in the past for their

general mental health concerns. Therefore, it is unclear whether our results would generalize to self-identified health-anxious individuals in the community who had not previously sought mental health treatment. We also did not ask our participants whether they sought treatment specifically for their health anxiety or for other mental health concerns in the past. In addition, our recruitment strategy required participants to self-identify with health anxiety, which may have introduced sampling bias and further limited the generalizability of our findings to the broader clinical population. The study was also only powered to detect large differences and underpowered to detect small-to-moderate between-group differences, meaning that we may not have detected differences of this size in our study. In addition, multiple comparison corrections were not applied to the analyses, as they were not specified prior to the analysis. Further, there were high levels of comorbidity between IAD and SSD, illustrating limitations of the current DSM-5. However, comorbidity between IAD and SSD was not reported at an individual level, as this was not the primary focus of the current study. High levels of comorbidity between SSD and IAD modified were also observed and should be considered when interpreting between-group comparisons. Finally, the cross-sectional nature of our study precludes conclusions about how IAD and SSD evolve and change over time. A longitudinal study would offer valuable insights into the progression and trajectories of these disorders.

## Clinical implications

The findings from this study have important implications for diagnosis, assessment and treatment of health anxiety. First, our results indicate that IAD is less common than SSD, suggesting that clinicians are more likely to encounter health-anxious individuals with moderate-to-severe somatic symptoms. Second, given the minimal differences observed between IAD and SSD, it may be appropriate for clinicians to consider a broader diagnosis for health-anxious individuals that is not constrained by somatic symptom severity. This would better reflect real-world clinical presentations, in which health anxiety can present across somatic symptom severity levels.

From a treatment perspective, cognitive–behavioral therapy (CBT) remains a well-supported intervention for health-anxious individuals [22], with past research showing that both IAD and SSD respond similarly to CBT [23]. Therefore, regardless of diagnostic label or somatic symptom severity, CBT appears to be an appropriate treatment option when health anxiety is present. However, for individuals with SSD who experience persistent and debilitating somatic symptoms in the absence of significant health anxiety (criterion B2), other interventions which target somatic symptom burden (i.e., drawn from the research into treatments for functional somatic symptoms) may be more appropriate [24, 25].

Finally, the findings from this study highlight the limitations of the categorical diagnostic model (i.e., DSM-5), which presumes discrete boundaries between disorders and fails to adequately capture the continuum of health anxiety presentations [26]. However, dimensional models, such as the Research Domain Criteria (RDoC) framework, offer a potential alternative by focusing on underlying neurobiological (e.g., brain functioning, genetics, emotional responses) and behavioral (e.g., family environment, social and cultural context) processes [27]. In the context of health anxiety, this approach would better capture the heterogeneity of health anxiety presentations and provide a more nuanced understanding of symptoms, such as excessive checking behaviors and online health searches, and persistent rumination about physical sensations and their meanings.

## Conclusions

Our findings contribute to the literature suggesting that among individuals experiencing severe or excessive health anxiety, there are minimal demographic or clinical differences between those who meet criteria for IAD and SSD. The differences that do exist seem to be a matter of severity [6–8]. Our novel findings also suggest that there are few differences between IAD with or without significant somatic symptoms. These findings call into question the validity and utility of arbitrarily splitting individuals with severe health anxiety into two separate diagnoses based solely on somatic symptom severity. Instead, a single diagnosis, such as IAD, may be more appropriate for health anxiety, as SSD remains a useful diagnostic category for individuals who are predominantly troubled by distressing somatic symptoms without experiencing

health anxiety (criterion B2). However, replication is needed along with longitudinal research following up individuals with IAD over time.

## Supporting information

**S1 Table. Demographic characteristics of the total sample.**
(DOCX)

**S2 Table. Nature and course of health anxiety of the total sample.**
(DOCX)

**S3 Table. Symptom severity, quality of life, and service utilization in the total sample.**
(DOCX)

**S4 Table. Health anxiety comorbidities with other mental health disorders and physical health conditions in the total sample.**
(DOCX)

**S5 Table. Demographic information of participants with current DSM-5 IAD and modified IAD diagnoses.**
(DOCX)

**S6 Table. Nature and course of participants with current DSM-5 IAD and modified IAD diagnoses.**
(DOCX)

**S7 Table. Symptom severity, quality of life, and service utilization of participants with current DSM-5 IAD and modified IAD diagnoses.**
(DOCX)

**S8 Table. Mental health and physical health comorbidities of participants with current DSM-5 IAD and modified IAD diagnoses.**
(DOCX)

**S9 Table. A summary of the key research findings.**
(DOCX)

## Acknowledgments

Thank you to Monique Holden (MH) for conducting the inter-rater reliability of the diagnostic interviews and to our participants who shared their experiences worrying about health.

## Author contributions

**Conceptualization:** Katarina Kikas, Aliza Werner-Seidler, Jill M. Newby.

**Data curation:** Katarina Kikas.

**Formal analysis:** Katarina Kikas.

**Funding acquisition:** Jill M. Newby.

**Investigation:** Katarina Kikas, Emily Upton, Brittany Corkish.

**Methodology:** Aliza Werner-Seidler, Jill M. Newby.

**Project administration:** Katarina Kikas, Aliza Werner-Seidler, Jill M. Newby.

**Resources:** Aliza Werner-Seidler, Jill M. Newby.

**Supervision:** Aliza Werner-Seidler, Jill M. Newby.

**Validation:** Aliza Werner-Seidler, Jill M. Newby.

**Visualization:** Katarina Kikas.

**Writing – original draft:** Katarina Kikas.

**Writing – review & editing:** Katarina Kikas, Emily Upton, Brittany Corkish, Aliza Werner-Seidler, Jill M. Newby.

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
