## [Decision Letter · Decision Letter 0]

23 Jul 2025

Dear Dr. Kikas,

Thank you for submitting your manuscript to PLOS ONE. After careful consideration, we feel that it has merit but does not fully meet PLOS ONE’s publication criteria as it currently stands. Therefore, we invite you to submit a revised version of the manuscript that addresses the points raised during the review process.

We look forward to receiving your revised manuscript.

Kind regards,

Omnia Hamdy, PhD

Academic Editor

PLOS ONE

Journal Requirements:

“Australian National Health and Medical Research Council Investigator grant (2008839)”

“Jill Newby receives grant funding from the NHMRC, Wellcome Trust, MRFF, NIHR, and KONESKI. Aliza Werner-Siedler receives grant funding from NHMRC and MRFF.”

Reviewers' comments:

Reviewer's Responses to Questions

**Comments to the Author**

1. Is the manuscript technically sound, and do the data support the conclusions?

Reviewer #1: Partly

Reviewer #2: Yes

2. Has the statistical analysis been performed appropriately and rigorously?

Reviewer #1: No

Reviewer #2: Yes

3. Have the authors made all data underlying the findings in their manuscript fully available?

Reviewer #1: Yes

Reviewer #2: Yes

4. Is the manuscript presented in an intelligible fashion and written in standard English?

Reviewer #1: No

Reviewer #2: Yes

Reviewer #1: Thank you for the opportunity to review your manuscript. I appreciate the effort that went into this study, which addresses a critical issue in the classification and understanding of health anxiety disorders. Your work provides valuable insights into the diagnostic overlap between Illness Anxiety Disorder (IAD) and Somatic Symptom Disorder (SSD), contributing to the ongoing discussion about the validity of DSM-5 distinctions in this area.

Overall, the study is methodologically sound and presents important findings; however, there are areas that require further clarification and refinement. While your results suggest minimal differences between IAD and SSD, the implications of this finding should be further explored. Additionally, there are some methodological details missing, and the presentation of results and discussion could be strengthened to better contextualize the findings in relation to previous research and clinical practice.

To enhance the clarity, rigor, and impact of the manuscript, I have outlined below several key revisions that should be addressed before the manuscript can be considered for publication.

Abstract:

Streamline verbose sentences for better readability. Ensure consistent verb tense and grammatical accuracy throughout the abstract. For ex. instead of "The objective of this paper was to compare IAD and SSD, and IAD to a modified IAD diagnosis (with moderate-to-severe somatic symptoms) in individuals experiencing health anxiety", rephrase as "This study compares IAD, SSD, and a modified IAD diagnosis (moderate-to-severe somatic symptoms) in health-anxious individuals.".

Introduction:

-Reduce redundancy in the background section: focus on the most relevant theoretical and empirical literature.

-Provide a more direct transition from the discussion of health anxiety to the study’s objectives.

-Expand on previous research that has challenged the DSM-5 distinctions between IAD and SSD and clearly explain how this study builds upon those findings.

Sampling and Recruitment:

-Clarify how participants were screened and recruited beyond self-identification as health-anxious individuals.

-Explain whether any efforts were made to ensure sample representativeness (e.g., gender, age, education).

Diagnostic Assessment and Self-Report Measures:

-Provide more details on interviewer training and how consistency was ensured in diagnostic assessments.

-Provide more details on blinding procedures: were assessors blind to the study hypotheses or participant group?

Data Analysis:

-Explain if multiple comparison corrections (e.g., Bonferroni) were applied to avoid Type I errors.

-Describe how missing data was handled in the analysis.

Results:

-Provide a clearer justification for the modified IAD category and whether it adds diagnostic value.

-Consider adding a summary table listing key findings to improve readability.

-Expand the discussion on healthcare utilization differences and their clinical relevance.

Discussion:

-Address why previous studies found greater differences between IAD and SSD and how your methodology may explain these discrepancies.

-Elaborate on clinical implications! If the distinction between IAD and SSD is minimal, what does this mean for diagnosis and treatment?

-Discuss alternative diagnostic models (e.g., a dimensional approach to health anxiety).

-Provide clearer treatment recommendations—Should IAD and SSD be approached similarly in therapy?

Limitations:

-Expand on potential selection bias. Self-identified health-anxious participants may not represent the full clinical population.

-Expand on cross-sectional design limitations. A longitudinal approach would provide deeper insights into the evolution of these disorders.

-Expand on reliance on self-report. Consider discussing how medical verification of symptoms could enhance future research.

General Points:

-Refine language and phrasing for clarity, particularly in the abstract and discussion sections.

-Ensure APA formatting for citations, references, and tables.

-Revise tables and figures to improve clarity and consistency in labeling and statistical reporting.

Reviewer #2: Introduction

The construct of modified IAD was based on the presence of somatic symptoms among people with IAD, which was highly assimilated the criteria of SSD. Such classification, without factor analysis from previous or current research, would be ambiguous in classifying people with the current purposed modified IAD or SSD. In addition, as the criteria of SSD itself in the DSM-5/DSM-5-TR includes the severity of somatic symptoms, regressors of somatic symptoms may inevitably correlate with SSD due to the natural definition of the disorder and induced biasness in the regression analysis.

The paragraph also did not specify the meaning of constructing a modified IAD from the current IAD vs SSD construct, neither theoretical nor clinical implications of such newly constructed classifications. It would require comparisons of the modified IAD with current existing SSD to illustrate such necessity of new classification.

Moreover, the flow of the introduction is recommended to be reorgnaized in a way to explain fully the current reserach rationale.

Measures, Statistical analyses

Line 182-203: by the construct of the modified IAD, are there any interviewee determined with moderate/severe somatic symptoms but not with SSD?

Results

The above questions were not reported nor addressed from the data comparison between IAD and SSD only, without the analyses between/among the newly constructed modified IAD. The question regarding the high similarity, if not congruence, of modified IAD with SSD, was not reported in the prevalence either.

The differences between current IAD and SSD were, as expected, due to the severity or presence of somatic symptoms, which was inborn in the definition of the disorders in the DSM-5/DSM-5-TR. Similar problem exists in the comparison between IAD and modified IAD, which brings biasedness in the regression analysis.

Sample sizes in some of the regressor groups within the regression analyses are obviously small and may be underpowered to demonstrate if there is a significant effects from the variables.

Exclusion of 11 participants who were in the group of "comorbid with IAD and SSD" was not explained.

Discussion, Limitation, Conclusions

Construct of modified IAD were not addressed in the latest sections, and the questions regarding the necessity of the new construct was not discussed. To demonstrate that the modified IAD exist, there may need a proof/research on whether the gaps, clinically or empirically, exist between IAD and SSD. For example, the consideration of Criterion F of IAD, the differences between modified IAD and SSD by factor analysis, etc.

Some of the rationale of the construct demonstrated tautology in the design and may need to be refined, e.g., the effects by severity of somatic symptoms which was within criteria of distinguishing IAD and SSD, the homogeneity of modified IAD and SSD, etc.

It would be interesting to see some of the participants comorbid with IAD and SSD, which should not be that case by their criteria (absence/mild vs presence/moderate or above somatic symptoms). The occurrence of such data may be worthy to research on whether the research gap exist within such classification.

It would be of interest to suggest for clinical implications of such classification and how different approaches may be suggested to handle the consequences from the different diagnoses.

**Do you want your identity to be public for this peer review?** For information about this choice, including consent withdrawal, please see our Privacy Policy

Reviewer #1: No

Reviewer #2: No

---

## [Author Response · Author response to Decision Letter 1]

4 Sep 2025

Reviewer 1

Comments to the author: Thank you for the opportunity to review your manuscript. I appreciate the effort that went into this study, which addresses a critical issue in the classification and understanding of health anxiety disorders. Your work provides valuable insights into the diagnostic overlap between Illness Anxiety Disorder (IAD) and Somatic Symptom Disorder (SSD), contributing to the ongoing discussion about the validity of DSM-5 distinctions in this area. Overall, the study is methodologically sound and presents important findings; however, there are areas that require further clarification and refinement. While your results suggest minimal differences between IAD and SSD, the implications of this finding should be further explored. Additionally, there are some methodological details missing, and the presentation of results and discussion could be strengthened to better contextualize the findings in relation to previous research and clinical practice.

To enhance the clarity, rigor, and impact of the manuscript, I have outlined below several key revisions that should be addressed before the manuscript can be considered for publication.

Response: We appreciate your positive and helpful feedback. We have carefully considered your feedback and revised our manuscript, as outlined in our responses below.

Abstract:

Comment: Streamline verbose sentences for better readability. Ensure consistent verb tense and grammatical accuracy throughout the abstract. For ex. instead of "The objective of this paper was to compare IAD and SSD, and IAD to a modified IAD diagnosis (with moderate-to-severe somatic symptoms) in individuals experiencing health anxiety", rephrase as "This study compares IAD, SSD, and a modified IAD diagnosis (moderate-to-severe somatic symptoms) in health-anxious individuals.".

Response: Thank you for your suggestion. We have amended the abstract on page 2 in line with these suggestions:

Abstract, page 2:

Little is known about the features and characteristics of illness anxiety disorder (IAD) and somatic symptom disorder (SSD), and whether the minimal somatic symptom requirement in IAD is valid and useful. This study compares IAD and SSD, and IAD (involving no or mild somatic symptoms) to a modified IAD diagnosis (modified to include moderate-to-severe somatic symptoms) in health anxious individuals. We recruited health anxious individuals drawn from the community and assessed IAD and SSD. We compared the disorders on demographic and clinical characteristics, health care utilization, and assessed the prevalence of IAD subtypes. The validity of the IAD somatic symptom criterion was examined by comparing current IAD with modified IAD. Among 118 participants, 28 met criteria for IAD, 47 for SSD, and 38 for modified IAD. Most with IAD fluctuated between seeking and avoiding medical care (71.4%), while 25% were care-seeking and 3.6% were care-avoidant. Demographic information, illness course, mental health comorbidities, and symptom severity did not differ significantly between IAD and SSD. However, individuals with SSD reported significantly more somatic symptoms, chronic health conditions, reduced quality of life, and more health care visits compared to individuals with IAD. Similar findings were observed between IAD-current and IAD-modified. The minimal differences between IAD and SSD, and between IAD-current and IAD-modified call into question the utility of distinguishing health anxiety presentations based on somatic symptom presence and severity. Further research should compare these disorders on illness course and treatment outcomes.

Introduction:

Comment: Reduce redundancy in the background section: focus on the most relevant theoretical and empirical literature. Provide a more direct transition from the discussion of health anxiety to the study’s objectives. Expand on previous research that has challenged the DSM-5 distinctions between IAD and SSD and clearly explain how this study builds upon those findings.

Response: Thank you for your comment. We have revised the introduction (pages 3-9) to improve readability, remove redundancy, focus on the most relevant empirical research, and provide a more direct link to our study objectives and prior work challenging the DSM-5 distinctions between IAD and SSD. We are happy to make further changes if required.

Sampling and Recruitment:

Comment: Clarify how participants were screened and recruited beyond self-identification as health-anxious individuals.

Response: Thank you for your question. The eligibility criterion was that individuals self-identified as being anxious or worried about their health or as having health anxiety. We intentionally kept this broad with minimal criteria to incorporate a diverse community-based sample of individuals who identified as worrying about their health, expanding on existing research which typically has involved treatment seeking samples only (e.g., Newby et al., 2017). To clarify this point, we have amended page 9 (Participants and Procedure) to provide additional detail on the targeted recruitment advertisements and screening process used to determine eligibility. This includes the specific recruitment messaging, the use of an online eligibility questionnaire, and the steps participants followed to take part in the study.

Methods, page 9:

One hundred and eighteen participants were recruited between November 2022 and 2023 through targeted online media advertisements on Instagram and Facebook. These advertisements were aimed at individuals who worried about their health or identified as having health anxiety, using messages such as “Are you suffering from health anxiety?” and “We are seeking research participants who experience persistent and frequent worries about their health…”. Interested individuals were directed to the Black Dog Institute website, which outlined the eligibility criteria and provided a link to a Qualtrics (2022) questionnaire. The questionnaire assessed eligibility, and eligible participants received an electronic participant information statement and consent form.

Comment: Explain whether any efforts were made to ensure sample representativeness (e.g., gender, age, education).

Response: The sample was not intended to be a representative sample of the general community, as the study specifically aimed to recruit a broad sample of individuals who self-identified as experiencing health anxiety. We have transparently reported the samples’ demographic characteristics in the Results section (Table 1; page 14) and in the Supporting Information section (S1 Table; page 34), and we have acknowledged the limitations regarding generalisability in the Discussion (page 27).

Discussion, page 27:

The present study has several limitations that need to be addressed. First, although participants were drawn from the community rather than as a treatment-seeking sample, the majority (90%) had sought treatment in the past for their general mental health concerns. Therefore, it is unclear whether our results would generalize to self-identified health-anxious individuals in the community who had not previously sought mental health treatment. We also did not ask our participants whether they sought treatment specifically for their health anxiety or for other mental health concerns in the past. In addition, our recruitment strategy required participants to self-identify with health anxiety, which may have introduced sampling bias and further limits the generalizability of our findings to the broader clinical population.

Diagnostic Assessment and Self-Report Measures

Comment: Provide more details on interviewer training and how consistency was ensured in diagnostic assessments.

Response: Thank you for your suggestion. We have included more information about interview training and how consistency was maintained in the Diagnostic Interview section on pages 11 and 12. We also conducted an inter-rater reliability assessment with two blinded assessors to evaluate reliability of diagnostic groups, noted on page 12.

Diagnostic Interview, pages 11 and 12:

The ADIS-V was administered by researchers KK and clinical psychologists EU and BC. EU and BC, already trained in administering the ADIS-V, provided training to KK. Consistency was maintained by adhering strictly to the structured interview format, using the exact wording of the script without deviation. We also assessed interrater reliability via blind ratings of audio recordings (see section below). Regular supervision with clinical psychologists and researchers AWS and JN was conducted to discuss any cases that required clarification, discussion and/or an additional rating.

Inter-rater reliability, page 12:

To assess inter-rater reliability, all interviews were recorded, and with participants meeting IAD or SSD criteria were assessed by an independent blinded rater. A subset of interviews were coded for the remaining diagnoses (e.g., MDD, 10%; n=18). Two clinical psychologists (BC, MH) listened to the recordings and made an independent diagnosis; these raters were blinded to the diagnosis recorded by the initial interviewer. Any discrepancies between the raters were resolved by a third rater (EU/JN). Kappa estimates between the two raters were 0.96 for IAD, 0.88 for SSD. Discrepancies for IAD and SSD arose from disagreements regarding the severity of somatic symptoms. Inter-rater agreement for other comorbid diagnoses was 100%, except for obsessive compulsive disorder (kappa=0.73).

Comment: Provide more details on blinding procedures: were assessors blind to the study hypotheses or participant group?

Response: The interviewers/assessors were not blind to the study hypotheses as they were responsible for determining participants’ diagnostic groups. However, for the inter-rater reliability assessment, two clinical psychologists were blinded to the participants’ diagnostic groups, which has been specified in the inter-rater reliability section (pages 12-13).

Inter-rater reliability, pages 12-13:

Two clinical psychologists (BC, MH) listened to the recordings and made an independent diagnosis; these raters were blinded to the diagnosis recorded by the initial interviewer.

Data Analysis:

Comment: Explain if multiple comparison corrections (e.g., Bonferroni) were applied to avoid Type I errors.

Response: Thank you for your question. Multiple comparison corrections, such as the Bonferroni method, were not applied to our analyses as these adjustments are intended for planned comparisons and should be applied in advance rather than post hoc (see paper below). We have also added a point about this in the limitations section (page 28).

- Hooper, R. (2025). To adjust, or not to adjust, for multiple comparisons. Journal of Clinical Epidemiology, 180. https://doi.org/10.1016/j.jclinepi.2025.111688

Limitations, page 28:

In addition, multiple comparison corrections were not applied to the analyses, as they were not specified prior to the analysis.

Comment: Describe how missing data was handled in the analysis.

Response: There were only two instances of missing data which occurred in the health care utilization section of the questionnaire (six questions in total). We retained these participants in the analysis and included all data available from them. For the health care utilization outcomes, the reported n (SSD n=45) reflects only those participants who provided responses to each question (see page 21).

Results

Comment: Provide a clearer justification for the modified IAD category and whether it adds diagnostic value.

Response: Thank you for your suggestion. We have made changes to the introduction (pages 7-9) to clarify the rationale for assessing the modified IAD group. Comparing current diagnostic criteria for IAD involving an absence or only mild somatic symptoms, to modified IAD allows us to test the validity of IAD criterion B (i.e., minimal somatic symptom criterion). This investigation allowed us to determine whether the somatic symptom criterion provides meaningful qualitative differences between two IAD presentations. In the discussion (page 27) we now reflect on diagnostic value of IAD, in line with the reviewer’s suggestion.

Introduction pages 7-8:

In the DSM-5, Criterion B for IAD specifies that health anxious individuals must have either no somatic symptoms or only mild ones, setting them apart from individuals who meet criteria for SSD who typically experience more pronounced physical symptoms. However, some researchers have questioned whether this strict IAD criteria is necessary and clinically useful [9], as clinical observations suggest that that some individuals with IAD present with moderate to severe somatic symptoms [9]. To date, no empirical studies have directly tested the validity of IAD Criterion B. Research is needed to compare individuals who meet current IAD criteria (i.e., no or minimal somatic symptoms) to those who meet all IAD criteria but report moderate-to-severe somatic symptoms. Comparing these groups is crucial for evaluating the validity of criterion B in the diagnosis of IAD. It will help to determine whether this criterion accurately captures how the disorder presents in real-world clinical settings or needs to be revised.

Introduction, pages 8-9:

The second aim of this study was to test the validity of IAD Criterion B, by examining whether health-anxious individuals who met current IAD criteria (i.e., no or mild somatic symptoms) differed from those who met all DSM-5 IAD criteria but reported experiencing moderate-to-severe somatic symptoms. We use the term ‘modified IAD diagnosis’ throughout this paper to describe the latter group.

Discussion, page 27:

This suggests that the presence or severity of somatic symptoms may be less important for clinicians when diagnosing an individual with IAD.

Comment: Consider adding a summary table listing key findings to improve readability.

Response: Thank you for the suggestion. We have included a summary table in our supporting information file (see S9 Table).

Comment: Expand the discussion on healthcare utilization differences and their clinical relevance.

Response: We have expanded our discussion of the clinical relevance of healthcare utilization differences in the discussion section (page 26). However, we emphasise that these inferences remain speculative.

Discussion, page 26:

These findings may suggest that higher somatic symptom severity in SSD may prompt individuals to seek more medical care than those with no or milder somatic symptoms, such as in IAD. Although speculative, interventions aimed at reducing or managing somatic symptoms could help reduce unnecessary healthcare visits, thereby lessening the burden on both the individual and the healthcare system.

Discussion

Comment: Address why previous studies found greater differences between IAD and SSD and how your methodology may explain these discrepancies.

Response: Thank you for your suggestion. As noted in the discussion (page 26), the key methodological difference between our study and previous research is the sample (treatment-seeking participants vs. participants drawn from the community) and this difference is likely to account for the discrepancies. We also acknowledge that other factors may have contributed to the discrepancies, but it is difficult to determine this with certainty.

Discussion, page 26:

The reasons underlying these discrepancies are unclear but may stem from sampling differences, as our study recruited health-anxious participants from the community, while past research recruited participants signing up to a treatment trial. It is possible that there may have been differences in severity or other factors which may have influenced the findings.

Comment: Elaborate on clinical implications! If the distinction between IAD and SSD is minimal, what does this mean for diagnosis and treatment? Discuss alternative diagnostic models (e.g., a dimensional approach to health anxiety). Provide clearer treatment recommendations—Should IAD and SSD be approached similarly in therapy?

Response: Thank you for this great suggestion. We have added a dedicated ‘Clinical implications’ section in the discussion (pages 29-30), in which we elaborate on the clinical implications of our findings, provide t

---

## [Decision Letter · Decision Letter 1]

26 Nov 2025

Dear Dr. Kikas,

Thank you for submitting your manuscript to PLOS ONE. After careful consideration, we feel that it has merit but does not fully meet PLOS ONE’s publication criteria as it currently stands. Therefore, we invite you to submit a revised version of the manuscript that addresses the points raised during the review process.

We look forward to receiving your revised manuscript.

Kind regards,

Mohammad Faezi Ghasemi, Ph.D

Academic Editor

PLOS ONE

Journal Requirements:

Reviewers' comments:

Reviewer's Responses to Questions

**Comments to the Author**

Reviewer #2: (No Response)

2. Is the manuscript technically sound, and do the data support the conclusions?

Reviewer #2: Partly

3. Has the statistical analysis been performed appropriately and rigorously?

Reviewer #2: N/A

4. Have the authors made all data underlying the findings in their manuscript fully available?

Reviewer #2: Yes

5. Is the manuscript presented in an intelligible fashion and written in standard English?

Reviewer #2: Yes

Reviewer #2: Thank you for the opportunity to review your manuscript. I appreciate the effort that went into the revision, however, there are some several previous comments not being addressed. In addition, there are some further comments (in revised manuscript with tracked changes) for the modified manuscript.

Introduction:

The revise on Criterion B of IAD has been suggested but not on Criterion F. It is still lacking of theoretical support of building the IAD modified when highly similar disorders (e.g., SSD) cover the spectrum with moderate/severe somatic symptoms. More framework of the necessity of such construct should be elaborated.

Lines 96-151

Overall, the passage presents a coherent summary of three studies examining the diagnostic validity and clinical utility of IAD and SSD. However, there are a few minor inconsistencies and interpretive tensions that warrant clarification. First, while Bailer et al. (2005) is described as examining IAD and SSD, the study appears to have used post-hoc operationalizations based on DSM-IV data rather than DSM-5 diagnostic criteria; this distinction should be made explicit to avoid implying that the DSM-5 disorders were directly assessed. Second, the synthesis of findings suggests that differences between IAD and SSD are largely a matter of severity rather than qualitative distinction, yet the Swedish study is noted to have found additional quantitative differences after controlling for health anxiety severity. This point slightly complicates the overall conclusion and could be acknowledged as such.

Method

Lines 284-287

The statistical result of inter-rater reliability shall be back to the Result section.

Previous Comment: Exclusion of 11 participants who were in the group of "comorbid with IAD and SSD" was not explained.

Thank you for your response for this comment. I do believe that your response should be included in the Method Section.

Limitation section:

Previous comment regarding possibilityof diagnostic overlap and comorbidity between modified IAD and SSD were not addessed in the limitation.

Previous Comment: Construct of modified IAD were not addressed in the latest sections, and the questions regarding the necessity of the new construct was not discussed. To demonstrate that the modified IAD exist, there may need a proof/research on whether the gaps, clinically or empirically, exist between IAD and SSD. For example, the consideration of Criterion F of IAD, the differences between

modified IAD and SSD by factor analysis, etc. Some of the rationale of the construct demonstrated

tautology in the design and may need to be refined, e.g., the effects by severity of somatic symptoms

which was within criteria of distinguishing IAD and SSD, the homogeneity of modified IAD and

SSD, etc.

Comment: Thank you for your clarification regarding this comment. However, even the Criterion F of IAD is not the major reserach question in the current study, this should be considered to be discussed in the limitation section and possibly for further research. Without this discussion, the statement from Discussion, page 28, "This suggests that the presence or severity of somatic symptoms may be less important for clinicians when diagnosing an individual with IAD." is not solid.

Previous Comment: It would be interesting to see some of the participants comorbid with IAD and SSD,

which should not be that case by their criteria (absence/mild vs presence/moderate or above somatic

symptoms). The occurrence of such data may be worthy to research on whether the research gap exist

within such classification.

Comment: Please include your response in the discussion/limitation section.

**Do you want your identity to be public for this peer review?** For information about this choice, including consent withdrawal, please see our Privacy Policy

Reviewer #2: No

---

## [Author Response · Author response to Decision Letter 2]

6 Jan 2026

Reviewer #2: Thank you for the opportunity to review your manuscript. I appreciate the effort that went into the revision, however, there are some several previous comments not being addressed. In addition, there are some further comments (in revised manuscript with tracked changes) for the modified manuscript.

Response: Thank you for your helpful suggestions in improving our manuscript.

Introduction

Comment: The revise on Criterion B of IAD has been suggested but not on Criterion F. It is still lacking of theoretical support of building the IAD modified when highly similar disorders (e.g., SSD) cover the spectrum with moderate/severe somatic symptoms. More framework of the necessity of such construct should be elaborated.

Response: Thank you for your comment. In the present paper, our focus was on Criterion B; however, we acknowledge that Criterion F is also relevant as it refers to differential diagnosis. While the DSM intends to distinguish diagnostic categories through Criterion F, a recognised limitation of its categorical approach is that most diagnoses share overlapping symptom profiles. Therefore, while Criterion F is important for differential diagnosis, our study focused on Criterion B to examine health anxious individuals with varying somatic symptom severity, whereas SSD is a broader construct that does not always encompass individuals with health anxiety.

Comment: (Lines 96-151) Overall, the passage presents a coherent summary of three studies examining the diagnostic validity and clinical utility of IAD and SSD. However, there are a few minor inconsistencies and interpretive tensions that warrant clarification. First, while Bailer et al. (2005) is described as examining IAD and SSD, the study appears to have used post-hoc operationalizations based on DSM-IV data rather than DSM-5 diagnostic criteria; this distinction should be made explicit to avoid implying that the DSM-5 disorders were directly assessed. Second, the synthesis of findings suggests that differences between IAD and SSD are largely a matter of severity rather than qualitative distinction, yet the Swedish study is noted to have found additional quantitative differences after controlling for health anxiety severity. This point slightly complicates the overall conclusion and could be acknowledged as such.

Response: Thank you for your suggestion. In our introduction we specified that the study by Bialer et al used a post-hoc operationalisation of DSM-IV IAD and SSD diagnoses on both lines 76 and 91-94.

Introduction, line 76:

“Bailer et al. [11] were the first to examine the clinical utility of distinguishing IAD and SSD on somatic symptoms using a ‘probable’ IAD and SSD diagnosis, which was operationalized from self-report questionnaires and clinical interviews.”

Introduction, lines 91-94:

“However, a limitation of this study was that IAD and SSD were operationalized using a post-hoc definition of the disorders, based on information gathered from self-report questionnaires and clinical interviews (for DSM-IV diagnoses) rather than being based on DSM-5 diagnostic criteria.”

Thank you for making your point about the Swedish study. We have now streamlined this section to align the conclusion with the additional findings.

Introduction, lines 104-116:

“More recently, a Swedish team aimed to replicate these results while accounting for trait health anxiety in a treatment-trial sample [12]. After controlling for trait health anxiety, they found no phenomenological differences in demographic and clinical characteristics related to health anxiety (e.g., age, gender, education, employment, anxiety sensitivity, intolerance to uncertainty, avoidance and safety behaviours), clinical course (e.g., age of onset) and visits to general practitioners or medical specialists. However, the authors found that SSD was associated with a greater number of psychologist visits, higher levels of disability, and more pain and tinnitus than IAD. These findings suggest that although minor differences between IAD and SSD remain after accounting for trait health anxiety, the overall distinction between the two diagnoses appear to have limited clinical value.”

Method

Comment: (Lines 284-287) The statistical result of inter-rater reliability shall be back to the Result section.

Response: Thank you for your suggestion. We have moved the inter-rater reliability results to the results section, please see page 15 (lines 263-267).

Previous Comment: Exclusion of 11 participants who were in the group of "comorbid with IAD and SSD" was not explained.

Current comment: Thank you for your response for this comment. I do believe that your response should be included in the Method Section.

Response: Thank you for your feedback. We have included this in our methods section under statistical analyses. See page 10:

“Participants meeting comorbid IAD and SSD criteria were excluded from analyses comparing IAD and SSD to address the research question of whether differences exist between the two disorders.

Limitation section

Comment: Previous comment regarding possibility of diagnostic overlap and comorbidity between modified IAD and SSD were not addressed in the limitation.

Response: We have now included this comment in our limitation section on page 26.

“High levels of comorbidity between SSD and IAD modified were also observed and should be considered when interpreting between-group comparisons.”

Previous Comment: Construct of modified IAD were not addressed in the latest sections, and the questions regarding the necessity of the new construct was not discussed. To demonstrate that the modified IAD exist, there may need a proof/research on whether the gaps, clinically or empirically, exist between IAD and SSD. For example, the consideration of Criterion F of IAD, the differences between modified IAD and SSD by factor analysis, etc. Some of the rationale of the construct demonstrated tautology in the design and may need to be refined, e.g., the effects by severity of somatic symptoms which was within criteria of distinguishing IAD and SSD, the homogeneity of modified IAD and SSD, etc.

Comment: Thank you for your clarification regarding this comment. However, even the Criterion F of IAD is not the major research question in the current study, this should be considered to be discussed in the limitation section and possibly for further research. Without this discussion, the statement from Discussion, page 28, "This suggests that the presence or severity of somatic symptoms may be less important for clinicians when diagnosing an individual with IAD." is not solid.

Response: Thank you for your suggestion. We have now considered Criterion F in our discussion section on page 26.

Discussion, page 26:

“More research is needed to examine the course and treatment response of IAD with or without somatic symptoms in various samples (i.e., treatment-seeking, primary and secondary care), as well as the role of differential DSM diagnostic criteria, such as Criterion F, in distinguishing IAD from related disorders including SSD.”

Previous Comment: It would be interesting to see some of the participants comorbid with IAD and SSD, which should not be that case by their criteria (absence/mild vs presence/moderate or above somatic symptoms). The occurrence of such data may be worthy to research on whether the research gap exist within such classification.

Comment: Please include your response in the discussion/limitation section.

Response: Thank you for your suggestion. We have discussed this limitation in the limitation section on page 26.

“Further, there were high levels of comorbidity between IAD and SSD, illustrating limitations of the current DSM-5. However, comorbidity between IAD and SSD was not reported at an individual level, as this was not the primary focus of the current study.”

---

## [Editor Report · Decision Letter 2]

25 Jan 2026

Illness anxiety disorder and somatic symptom disorder: Similarities and differences in health-anxious individuals

PONE-D-24-57393R2

Dear Dr. Newby,

We’re pleased to inform you that your manuscript has been judged scientifically suitable for publication and will be formally accepted for publication once it meets all outstanding technical requirements.

Kind regards,

Mohammad Faezi Ghasemi, Ph.D

Academic Editor

PLOS One
---

## [Editor Report · Acceptance letter]

PONE-D-24-57393R2

PLOS One

Dear Dr. Newby,

I'm pleased to inform you that your manuscript has been deemed suitable for publication in PLOS One. Congratulations! Your manuscript is now being handed over to our production team.

Kind regards,

on behalf of

Dr. Mohammad Faezi Ghasemi

Academic Editor

PLOS One